# Mutations in fibulin-1 and collagen IV suppress the short healthspan of *mig-17/ ADAMTS* mutants in *Caenorhabditis elegans*

**Yukimasa Shibata❓\*, Yijing Huang, Moeka Yoshida, Kiyoji Nishiwaki**

Department of Bioscience, Kwansei Gakuin University, Sanda, Japan

\* yukshibata@kwansei.ac.jp

**Data Availability Statement:** All relevant data are within the manuscript and its Supporting information files.

## Abstract

The ADAMTS (a disintegrin and metalloprotease with thrombospondin motifs) family metalloprotease MIG-17 plays a crucial role in the migration of gonadal distal tip cells (DTCs) in *Caenorhabditis elegans*. MIG-17 is secreted from the body wall muscle cells and localizes to the basement membranes (BMs) of various tissues including the gonadal BM where it regulates DTC migration through its catalytic activity. Missense mutations in the BM protein genes, *let-2/collagen IV a2* and *fbl-1/fibulin-1*, have been identified as suppressors of the gonadal defects observed in *mig-17* mutants. Genetic analyses indicate that LET-2 and FBL-1 act downstream of MIG-17 to regulate DTC migration. In addition to the control of DTC migration, MIG-17 also plays a role in healthspan, but not in lifespan. Here, we examined whether *let-2* and *fbl-1* alleles can suppress the age-related phenotypes of *mig-17* mutants. *let-2(k196)* fully and *fbl-1(k201)* partly, but not *let-2(k193)* and *fbl-1(k206)*, suppressed the senescence defects of *mig-17*. Interestingly, *fbl-1(k206)*, but not *fbl-1(k201)* or *let-2* alleles, exhibited an extended lifespan compared to the wild type when combined with *mig-17*. These results reveal allele specific interactions between *let-2* or *fbl-1* and *mig-17* in age-related phenotypes, indicating that basement membrane physiology plays an important role in organismal aging.

## Introduction

The BMs are complex structures composed of multiple types of molecules. To maintain their integrity, it is necessary to appropriately remove damaged components and incorporate new ones, ensuring proper turnover. However, as animals age, the metabolism of BMs slows down, leading to the accumulation of damage [1]. Collagen levels decrease with age, while spontaneous cross-links increase. Yet, the anti-aging effects of BMs remain poorly understood. In the skin, BM collagen is essential for stem cell maintenance through hemidesmosomes [2]. Loss of collagen is known to impair cell competition, thus contributing to aging. However, the relationship between BMs and the suppression of aging beyond skin stem cells remains unknown.

The ADAMTS family of zinc metalloproteases is a group of enzymes that play important roles in various biological processes by cleaving proteins in the extracellular matrix (ECM) and

**Funding:** Grant-in-Aid for Research Activity Start-up by the Ministry of Education, Culture, Sports, Science and Technology to YS(22K20658) and by the Naito Grant for the advancement of natural science to KN. The funders had no role in study design, data collection and analysis, decision to publish, or preparation of the manuscript.

**Competing interests:** The authors have declared that no competing interests exist.

regulating the structure and function of tissues in the body [3]. Dysregulation of ADAMTS enzymes can contribute to various diseases and disorders, including arthritis, cardiovascular diseases, and cancer [4]. Although the functions of ADAMTS proteases in animal development and pathologies are well studied, their roles in organismal aging and lifespan remain mostly unsolved. In this study, we analyzed the functions of one of the *C. elegans* ADAMTS enzymes, MIG-17, in aging and lifespan.

MIG-17 is secreted from the body wall muscle cells as a proform and localizes to the BMs of various tissues, including gonad, intestine, body wall muscle, and hypodermis, where it is activated by auto-catalytic removal of its prodomain [5, 6] (Fig 1A and 1B). The protease activity of MIG-17 in the BM of DTCs is sufficient to regulate the directional migration of DTCs to generate the U-shaped gonad arms. In *mig-17* mutants, the DTCs meander and stray, resulting in an abnormal gonadal shape [7]. Dominant gain-of-function (*gf*) mutations in the BM molecules collagen IV α2 chain and fibulin-1 suppress the gonadal defect of the *mig-17* mutants [8–10] (Fig 1A). *let-2(gf)* mutations are amino-acid substitutions either in the triple helical domain or in the C-terminal non-collagenous 1 (NC1) domain (Fig 1A). *fbl-1(gf)* mutations are amino-acid substitutions found in the second EGF-like motif (Fig 1A). Genetic analyses revealed a regulatory pathway in which MIG-17 recruits and activates FBL-1C (C isoform), which then recruits NID-1 to the BM to control DTC migration. MIG-17 also activates collagen IV to induce nidogen-dependent and -independent pathways to control DTC migration [9]. In addition to its role in in gonadogenesis, MIG-17 also functions in regulating healthy aging. *mig-17* mutants exhibit an earlier decline in periodic behaviors, including the defecation cycle and pumping rate, as well as a decrease in motility rate, and show earlier shortening of body length compared to the wild type. [11].

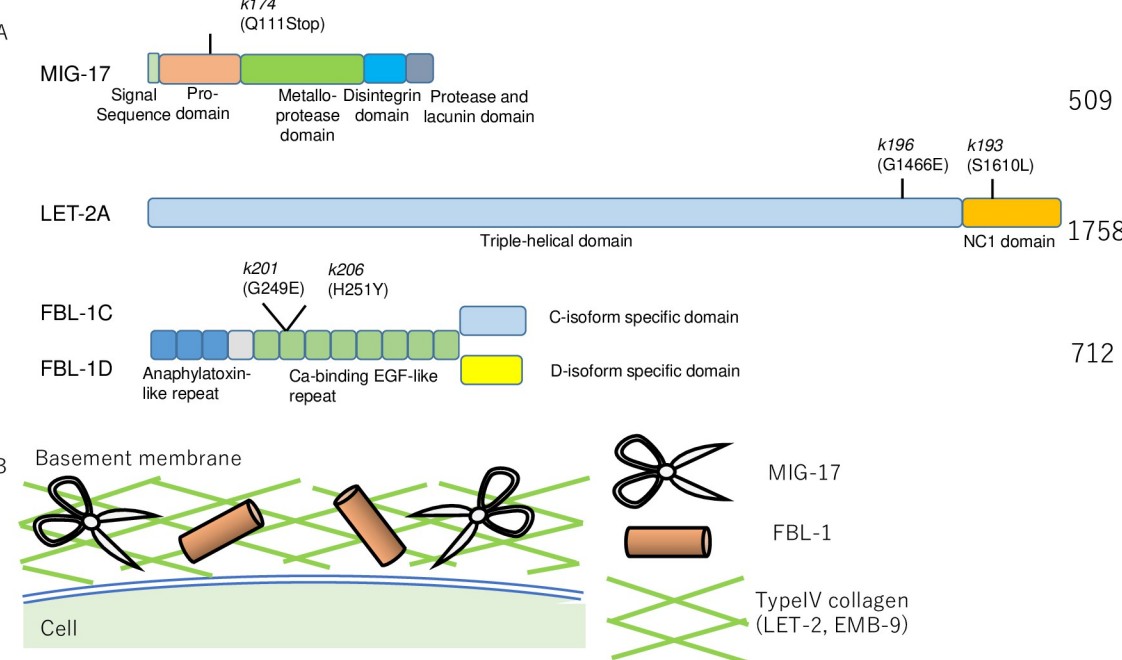

**Fig 1. Schematic diagrams of MIG-17, FBL-1C, and LET-2A.** (A) The mutation sites of *mig-17(k174)*, *let-2(k193, k196)*, and *fbl-1 (k201, k206)* are indicated. (B) A diagram showing the localization of MIG-17, LET-2, and FBL-1. These proteins are components of BMs.

In the present study, we examined the effects of suppressor mutations *let-2* and *fbl-1* on the age-dependent phenotypes of *mig-17* mutants. We found that *let-2* and *fbl-1* mutations suppressed the *mig-17* defects in an allele-specific manner.

## Materials and methods

### Strains and genetic analysis

Culture and handling of *C. elegans* were conducted as described [12]. Worms were cultured at 20˚C. The following mutations and transgenes were used in this work: *mig-17(k174)*, *fbl-1 (k201, k206)*, *let-2 (k193, k196)* [7–9].

### Microscopy

Gonad migration phenotypes were scored using a DIC images of Nomarski microscope (Axioplan 2; Zeiss). Gonad migration phenotypes were scored using DIC images captured with a Zeiss Axioplan 2 microscope [13].

### Analysis of growth rate

The growth rate was analyzed as described [14]. Briefly, newly hatched larvae were synchronized for one hour, grown at 20˚C, and the growth rate was assessed based on the stages of vulval development at the L4 stage. A simple invagination occurs in early L4, followed by a Christmas tree-like invagination in mid-L4, and the invagination becomes mostly closed in late L4.

### Analysis of body length

Body length was analyzed as previously described [11]. Briefly, young adult animals were collected as day 1 adults for synchronization. Body length was measured by ImageJ software. To measure body length, we traced a line along the body using ImageJ and then measured its length.

### Behavioral analyses

Defecation cycle and pumping rate were analyzed as previously described [11]. Briefly, average of five defecation cycles were scored for individual animal. If the defecation cycle exceeded 3 minutes, further measurements were discontinued. Number of pumps during 30 seconds were counted and the average of three independent measurements was calculated for each individual.

### Analysis of lifespan

Lifespan was analyzed as previously described [11]. Three independent experiments are performed. Synchronization was performed by collecting L4 animals. Animals were transferred to new OP50 seeded plates for every two days during reproduction period, and every three days during post-reproduction period.

### Statistics

Growth rates were analyzed using Fisher's exact test to compare adults and larvae.

One-way analysis of variance (ANOVA) was conducted using Excel to compare defecation cycle, pumping rate, and body length. If a significant difference was found by ANOVA, Tukey's multiple comparison test was performed using Excel statistics [15].

The lifespan was analyzed using a log-rank test in R [11].

## Results

### *fbl-1(k201)* mutation suppresses growth retardation of *mig-17* mutants

We previously reported that dominant gain-of-function mutations in *let-2* and *fbl-1* could suppress the gonadal defects of *mig-17* mutants (S1 Fig). We examined whether *mig-17* and the suppressor mutations affect the larval growth rate. The growth of *C. elegans* from hatched larvae to adults involves four larval stages, namely larval stage 1 (L1) to L4, punctuated by molting. We first examined whether MIG-17 is required for normal growth rate. We synchronized the newly hatched larvae for one hour and grew them at 20°C. The growth rate was assessed by monitoring the stages of vulval development at the L4 stage, as described [14]. Our observations revealed that *mig-17* mutants exhibited slower growth compared to wild-type animals (Fig 2A–2D). Since the gain-of-function mutations in collagen IV, *let-2(k193)* and *let-2(k196)*, and fibulin-1, *fbl-1(k201)* and *fbl-1(k206)* suppress the gonadal defects of *mig-17* mutants [8, 10], we further investigated whether they could also alleviate the growth retardation observed in *mig-17* mutants. *let-2(k193)* and *let-2(k196)* mutants also displayed growth retardation (WT vs *let-2(k196)* p = 0.058). These *let-2* mutations failed to suppress the slow growth observed in *mig-17* mutants. Instead, *let-2(k193)* seemed to exacerbate the phenotype of *mig-17* at 43 hours, as evidenced by the presence of 10% early L4 animals, a feature not observed in *mig-17* alone (Fig 2A and 2B). The mutations *fbl-1(k201)* and *fbl-1(k206)* did not significantly impact the growth rate, and only *fbl-1(k201)* exhibited a weak suppression of *mig-17* growth

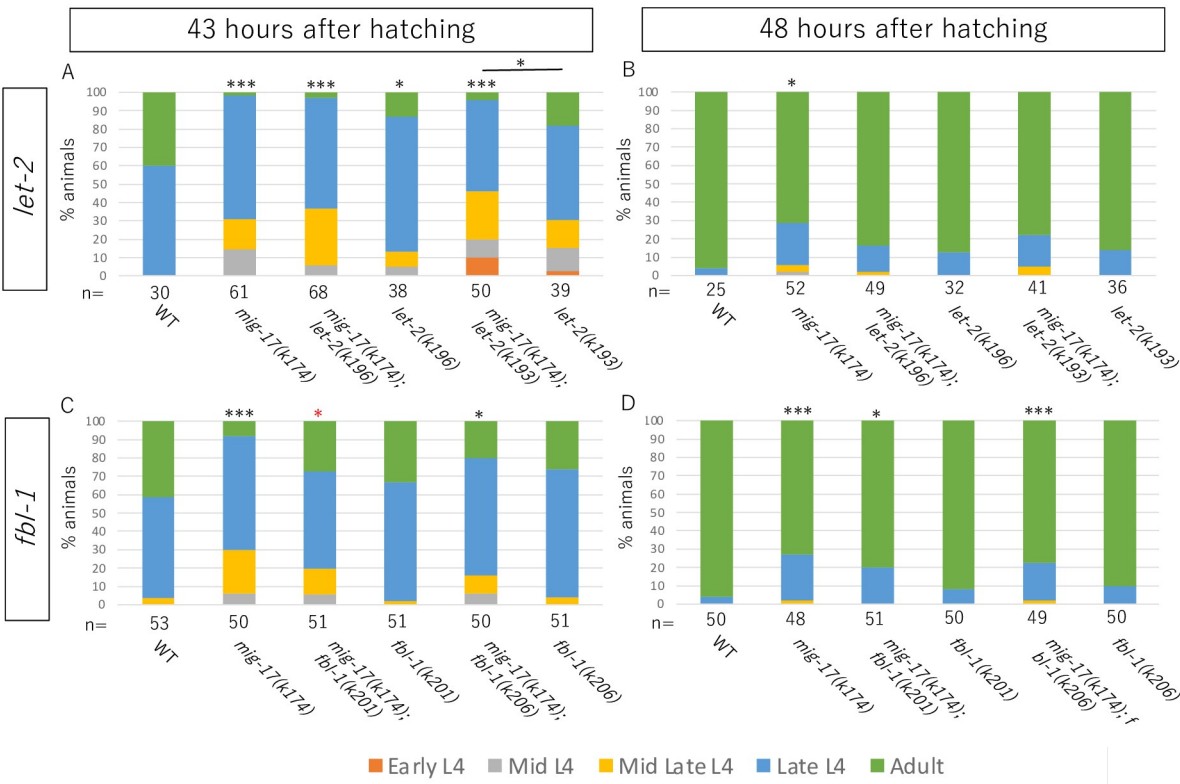

**Fig 2. Suppressors of the gonadal defects of *mig-17* fail to suppress the growth defect.** Bar graphs represent developmental stages at 43 or 48 hours after hatch. The sample size is indicated at the bottom of each bar graph. Percentages of adult animals and those of L4 animals were compared in 43 and 48 hours, respectively. Black and red asterisks on the top of bar graph indicate *p*-values for Fisher's exact test against WT and *mig-17(k174)*, respectively: ***p < 0.005, *p < 0.05.

retardation at 43 hours (Fig 2C). These results indicate that although both the *let-2* and *fbl-1* alleles are strong suppressors of the gonadal defects of *mig-17*, they mostly fail to suppress the growth defect of *mig-17*.

## Suppression of short healthspan in *mig-17* mutants

Aged decline in the defecation cycle and pumping rate are often used as indicators of senescence rate in *C. elegans* [11, 16, 17]. We examined the defecation cycle of wild type and *mig-17* mutant animals. The cycle was around 50 seconds at the young adult stage (day 1) and extended to 90 seconds until day 5 in the wild type (Fig 3A–3F). Although *mig-17* mutants showed a cycle comparable to the wild type at day 1, it became significantly longer after day 3 compared to the wild type, suggesting that the senescence rate of *mig-17* mutants is faster than that of the wild type (Fig 3A–3F) [11]. We then investigated whether the *let-2* and *fbl-1* mutants could suppress the *mig-17* defect. Both *let-2* alleles showed a significantly longer defecation cycle compared to the wild type from day 1 to day 5. Interestingly, however, *let-2(k196)*, but not *let-2(k193)*, strongly suppressed the *mig-17* defecation decline at day 3 and day 5 (Fig 3A–3C). Thus, *mig-17(k174)* and *let-2(k196)* can suppress each other for their defecation defects. *fbl-1(k206)* alleles showed significantly longer defecation cycles compared to the wild type after day 5 and *fbl-1(k201)* showed a trend toward suppression of the *mig-17* defecation decline at day 5, although it was not statistically significant (t-test: *p* = 0.069) (Fig 3D–3F).

We examined pharyngeal pumping rates at day 1 and day 9 adults. The pumping rates decreased by 50% in day 9 adults. The pumping rates of *mig-17* adults were much slower than those of wild type at day 1 and day 9, suggesting short healthspan of *mig-17* animals

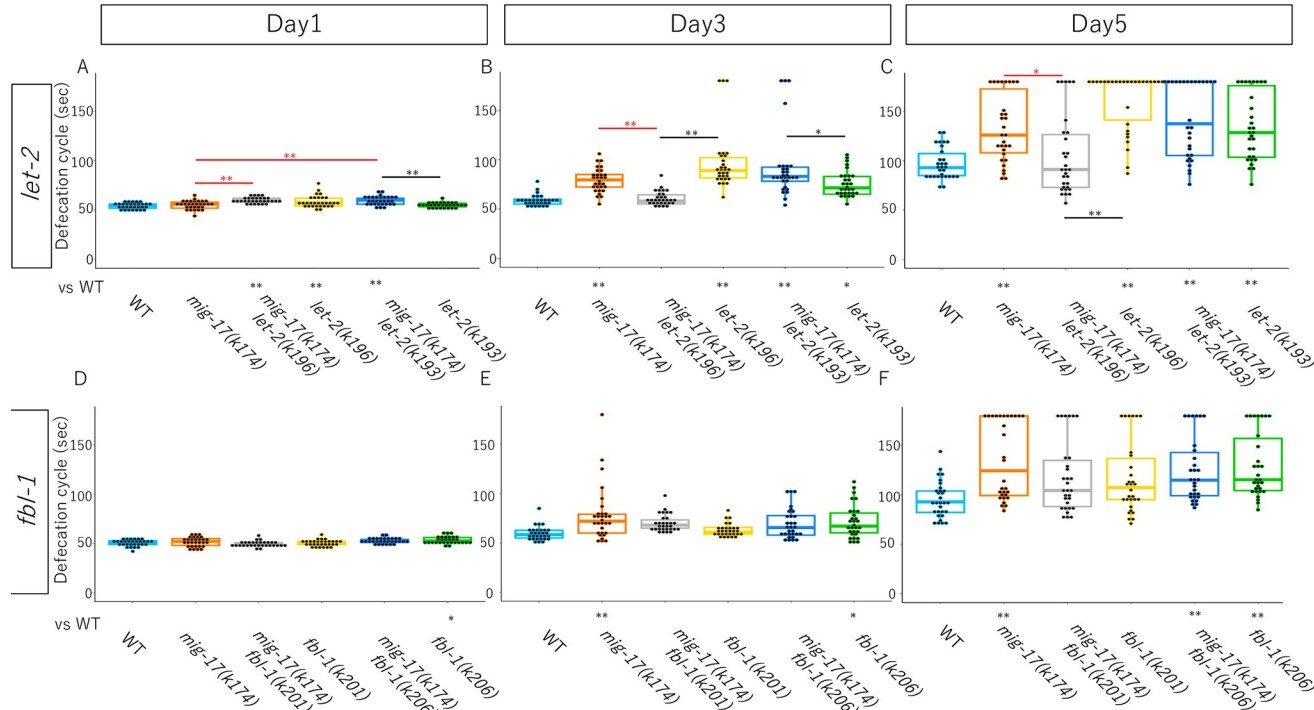

**Fig 3. Suppression of slow defecation cycle in aged *mig-17* mutants by *let-2(k196)*.** Box and dot plots indicate the defecation cycle. The sample size is 30 animals. *p*-values against WT is indicate at the bottom of plots. *p*-values against *mig-17(k174)* are indicated by red asterisks. One-way ANOVA with Tukey's multiple comparisons: *\*p*≤ 0.05, *\*\*p*≤ 0.01.

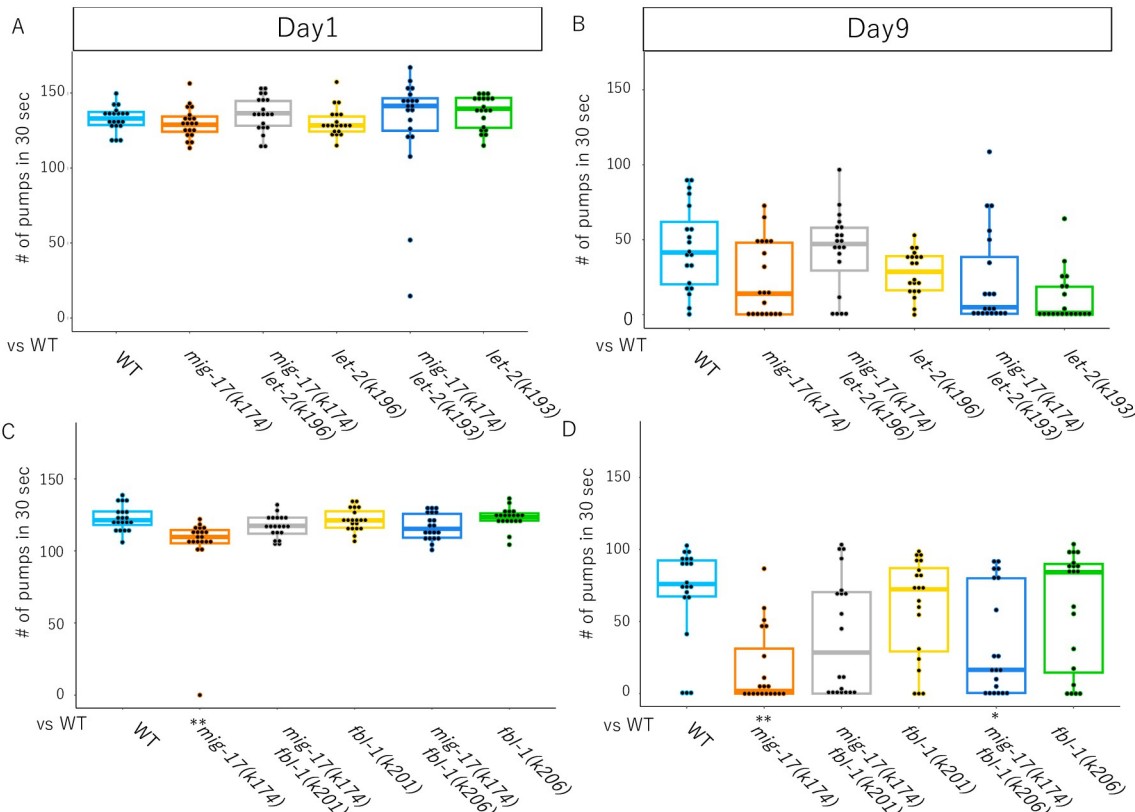

**Fig 4. Suppression of slow pumping rate in aged *mig-17* mutants by *let-2(k196)* and *fbl-1(k201)*.** Box and dot plots indicate the number of pumping in 30 seconds. Sample size is 20 animals. *p*-values against WT is indicate at the bottom of plots. *p*-values against *mig-17(k174)* are indicated by red asterisks. One-way ANOVA with Tukey's multiple comparisons: *$p \leq$ 0.05, **$p \leq$ 0.01.

(Fig 4C and 4D) [11]. Interestingly, *let-2(k196)*, but not *let-2(k193)*, showed a trend toward suppression of the *mig-17* pumping rate decline at day 9 (Fig 4A and 4B), although it was not statistically significant. The pumping rate of *mig-17(k174)* (*p* = 0.0001) and *fbl-1(k206) mig-17(k174)* (*p* = 0.028), but not that of *fbl-1(k201); mig-17(k174)* (*p* = 0.105), showed significant differences from that of wild-type animals at day 9 adult (Fig 4C and 4D).

We then examined age-dependent alteration of body length. In wild-type animals, the body length increased until day 3 to day 5 and gradually deceased after day 7 (Fig 5A and 5B). Although *mig-17* animals showed body length similar to that of the wild type at day1, their body lengths at day 3 and day 5 were significantly shorter than those of the wild type (Fig 5A and 5B) [11]. The body length of day 1 adults was longer in *mig-17; let-2(k196)* and *mig-17; let-2(k193)* double mutants compared to the wild type or *mig-17* mutants. At the day3 adult, the body length of *mig-17(k174)* and *let-2(k193) mig-17(k174)*, but not that of *let-2(k196) mig-17(k174)*, showed significant differences from those of wild-type animals (Fig 5A). Both *fbl-1(k201)* and *fbl-1(k206)* suppressed the short body size of *mig-17* at day 1 to day 3 (Fig 5B).

## The *fbl-1(k206)* mutation extends the lifespan of *mig-17* mutants

We examined the lifespan of *mig-17* mutants and analyzed the effects of *let-2* and *fbl-1* mutations on lifespan. The lifespan analysis was conducted from the young adult stage. The *mig-17* mutants exhibited a lifespan comparable to the wild type (Fig 6A–6D) as we have reported

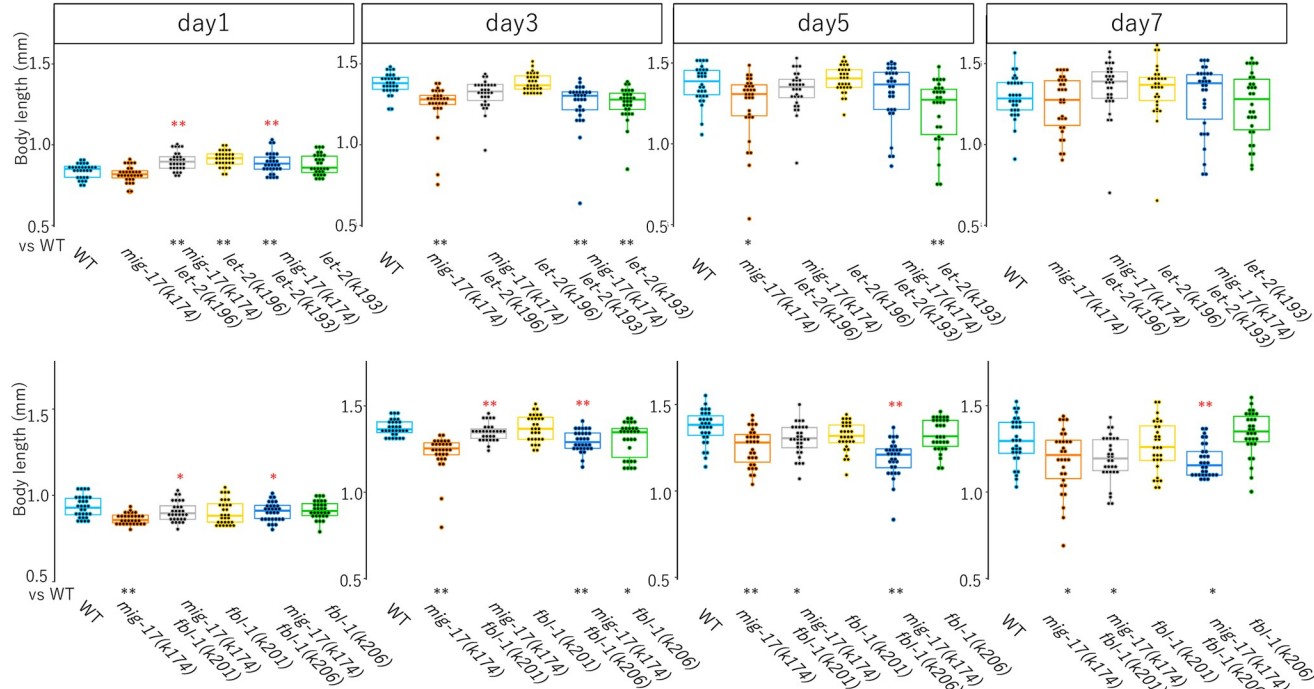

**Fig 5. Suppression of body length defects in *mig-17* mutants by *let-2* and *fbl-1*.** Box and dot plots indicate the body length. Standard deviation is indicated. The sample size is 30 animals. *p*-values against WT are indicated at the bottom of the plots. Red and blue asterisks on the top of bar indicate *p*-values against *mig-17(k174)* and *let-2* or *fbl-1*, respectively. One-way ANOVA with Tukey's multiple comparisons: *$p \leq$ 0.05, **$p \leq$ 0.01.

[11]. However, the lifespans of *let-2(k193)* and *let-2(k196)* animals were significantly shorter than the wild type or *mig-17* mutants. The *let-2(k193); mig-17* and *let-2(k196); mig-17* double mutants also showed a shortened lifespan (Fig 6A, 6B and 6E). In contrast, the lifespans of *fbl-1(k201)* and *fbl-1(k206)* single mutants, as well as *fbl-1(k201); mig-17* double mutants, were comparable to those of the wild type or *mig-17* mutants (Fig 6C–6E). Interestingly, however, *fbl-1(k206); mig-17* double mutants exhibited a significantly longer lifespan compared to the wild type. Thus, as observed in the age-related phenotypes, *let-2* and *fbl-1* mutations differentially affected the lifespan of *mig-17* mutants. The extension of lifespan in *fbl-1(k206); mig-17* double mutants, but not in *fbl-1(k206)* single mutants, implies that *fbl-1(k206)* mutation can extend lifespan in the absence of the MIG-17 protease (since *mig-17(k174)* represents a null allele).

## Discussion

In the present study, we analyzed the age-related phenotypes of the *mig-17/ADAMTS* mutant and its suppressor mutations for the gonadal defects in *let-2/collagen IV* and *fbl-1/fibulin-1*. The *mig-17(k174)* null mutant showed a normal lifespan but exhibited a slow growth rate and short healthspan, including earlier elongation of defecation cycle and a reduction of pharyngeal pumping rate. The *mig-17* mutant also exhibited a defect in age-associated body length elongation. Despite all *let-2* and *fbl-1* alleles being robust suppressors of the *mig-17* gonadal defects, they had differential effects on these *mig-17* phenotypes.

As we observed in *mig-17/ADAMTS* mutants, growth retardation was also noted in *Adamts1* as well as *Adamts9/+; Adamts20* mutants in mice [18, 19]. Although the *mig-17*

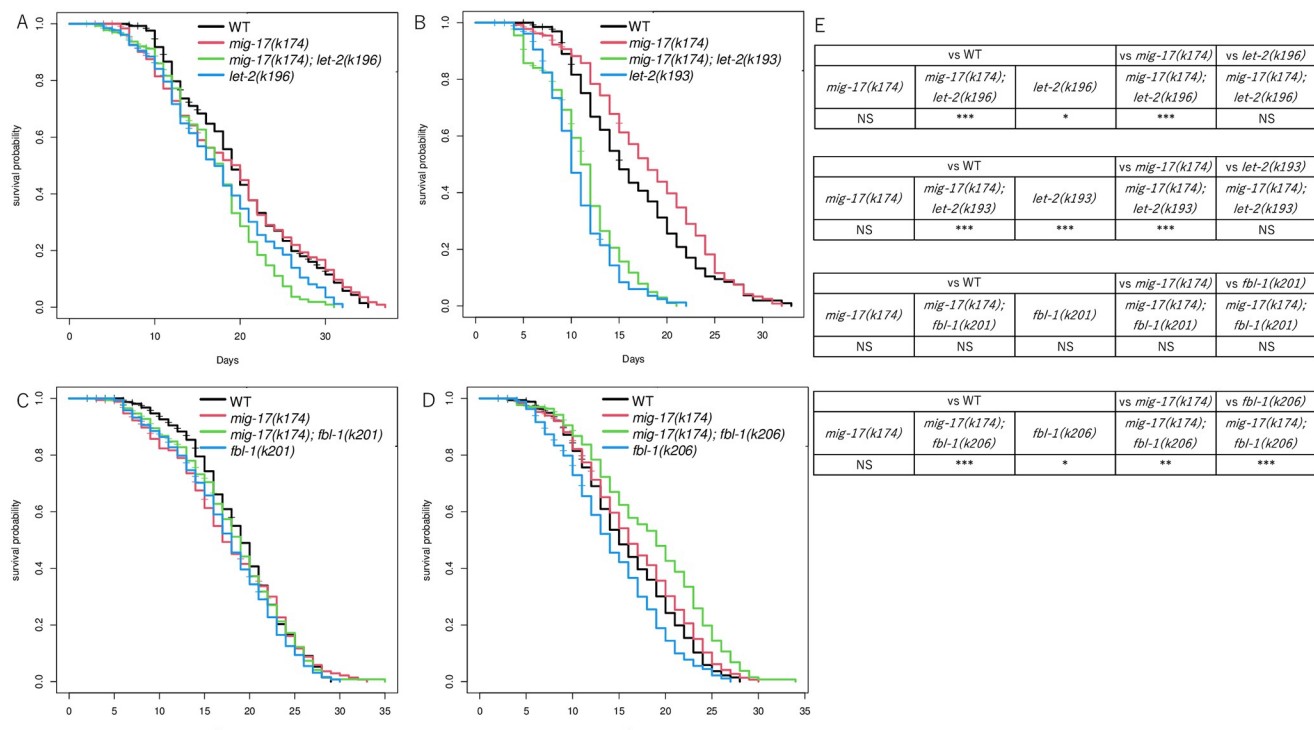

**Fig 6. Lifespan extension by *mig-22(k185gf)*.** (A-D) Survival curve shows the effect of *let-2(k193, k196)* or *fbl-1(k201, k206)* mutations in the *mig-17(k174)* background. The x axis represents lifespan in days of adulthood. The y axis shows the fraction of worms alive. (E) Table shows *p*-values for logrank test are indicated as: \*\*\**p* < 0.005, \*\**p* < 0.01, \**p* < 0.05, NS Not significant. The sample size is 180 animals.

mutants had no effect on lifespan, they exhibited phenotypes associated with short healthspan. While much of the research on ADAMTS proteases has focused on their roles in development and disease, some studies suggest that they may also have functions relevant to aging. For example, several ADAMTS proteases are induced in intervertebral disc degeneration during aging [20, 21]. Reduction of the reelin level by ADAMTS-4 and ADAMTS-5-dependent degradation is associated with hyperphosphorylated tau protein, forming neurofibrillary tangles leading to Alzheimer's disease [22]. However, these represent deleterious effects of ADAMTS on aging. In contrast, MIG-17 has a function to promote healthy aging, suggesting that MIG-17-dependent remodeling of the basement membrane is crucial for delaying the senescence process. We previously demonstrated that chondroitin proteoglycan acts downstream of MIG-17 to mediate healthy aging [11].

Both *let-2* mutations shortened lifespan. Thus, both of these type IV collagen mutations have a negative impact on lifespan, although it is not clear whether these mutations directly affect lifespan. However, *let-2(k196)* suppressed the early prolongation of *mig-17* defecation cycle associated with aging (Table 1). *let-2(k196)* also suppressed *mig-17* for its insufficient increase in body length with aging. In contrast, *let-2(k193)* did not suppress any of these aging defects in *mig-17*. Type IV collagen molecules self-assemble into a complex mesh-like network within basement membranes. This network contributes to the mechanical integrity and resilience of basement membranes, allowing them to withstand mechanical stresses [23]. In addition to these structural functions, type IV collagen is also known to be involved in TGF-β signaling. For example, mutations in *COL4A1* and *COL4A2* impair triple helical assembly into protomers and their secretion, leading to intracellular accumulation and

**Table 1.**

| Phenotype | Developmental phenotype | | Aging | | | |
|---|---|---|---|---|---|---|
| | Gonad development | Growth rate | Defecation cycle | Pumping rate | Body length | Lifespan |
| *mig-17(k174)* | Mig | Slow | Slow | Slow | Short | |
| *mig-17(k174) let-2(k196)* | | Slow | | | | Short |
| *mig-17(k174) let-2(k193)* | | Slow | Slow | | Short | Short |
| *let-2(k196)* | | Slow | Slow | | | Short |
| *let-2(k193)* | | | Slow | | Short | Short |
| *mig-17(k174) fbl-1(k201)* | | | | | | |
| *mig-17(k174) fbl-1(k206)* | | Slow | Slow | Slow | Short | Long |
| *fbl-1(k201)* | | | | | | |
| *fbl-1(k206)* | | | Slow | | Short | Short |

extracellular deficiency. Accordingly, semi-dominant *COL4A1* and *COL4A2* mutations cause Gould syndrome characterized primary by cerebrovascular manifestation in which TGF-β signaling is overactivated [24]. Increased TGF-β1 and TGF-β-dependent SMAD3 signaling are associated with age-related aortic valve calcification in *klotho* knockout mice [25]. Interestingly, DAF-7/TGF-β regulates longevity through DAF-2/insulin signaling in *C. elegans* [26]. It may be possible that the *let-2(k196)* mutation alters the TGF-β signaling to correct the short healthspan of *mig-17* mutants. Consistently, in humans, ADAMTS10, whose mutation causes Weill-Marchesani syndrome characterized by short stature and thickened skin, increases TGF-β activation in a dose-dependent manner [27, 28]. *let-2(k196)* is a missense mutation in the triple helical domain, whereas *let-2(k193)* is a missense mutation in the C-terminal NC1 domain of type IV collagen [9]. It might be possible that the triple helical domain may play a more important role than the NC1 domain in the regulation of aging.

*fbl-1(k201)*, but not *fbl-1(k206)*, suppressed the growth retardation and early decline in pumping rate of *mig-17*. Furthermore, both *fbl-1* alleles suppressed the insufficient growth of body length associated with aging in *mig-17*. *fbl-1(k201)* had no effect on lifespan, but *fbl-1 (k206)* had a slightly shorter lifespan than the wild type. However, surprisingly, the *mig-17 (k174); fbl-1(k206)* double mutant lived significantly longer than the wild type animals. Interestingly, *fbl-1(k201)* and *fbl-1(k206)* mutations are amino acid substitutions in the second Calcium binding EGF-like motif (G249E and H251Y, respectively). Thus, they are only separated by one amino acid within the same EGF-like motif. It is surprising that only *fbl-1 (k206)*, but not *fbl-1(k201)*, interacts with *mig-17* to prolong lifespan. Fibulin-1 is an evolutionarily conserved basement membrane protein that provides structural support to cells and tissues [29]. Fibulin-1 is also known to regulate EGF signaling. Fibulin-1C and D isoforms bind to the EGF receptor, possibly through its EGF-like motifs, and inhibits its activation, localization and function of lung cancer cells [30]. In *C. elegans*, the LIN-3/EGF and LET-23/EGF receptor signaling promotes healthy aging and longevity [31]. Thus, it might be possible that FBL-1 regulates lifespan by modulating EGF signaling in *C. elegans*. Among two splicing isoforms, FBL-1C and FBL-1D, FBL-1C is essential for the suppression of *mig-17* gonadal defects [8]. It is unknown which isoform acts in the suppression of age-related phenotypes of *mig-17*. It would be interesting to determine which isoform, or both, is required for the suppression of age-related phenotypes of *mig-17* and whether they act in signaling pathways involving EGF.

## Supporting information

**S1 Fig. Suppression of the gonad migration defect of *mig-17* by *let-2* or *fbl-1*.** Percentage of DTC migration defects. Blue and orange bars represent defects in anterior and posterior gonad arms, respectively. Black and red asterisks indicate *p*-values for Fisher's exact test against WT and *mig-17(k174)*, respectively *p*-values for Fisher's exact test are indicated: ***$p < 0.005$, *$p < 0.05$.
(TIF)

## Acknowledgments

We thank Noriko Nakagawa, Nami Okahashi, and Chizu Yoshikata for technical assistance. Some nematode strains used in this work were provided by the Caenorhabditis Genetics Center.

## Author Contributions

**Conceptualization:** Yukimasa Shibata, Kiyoji Nishiwaki.

**Formal analysis:** Yijing Huang, Moeka Yoshida.

**Funding acquisition:** Yukimasa Shibata, Kiyoji Nishiwaki.

**Investigation:** Yukimasa Shibata.

**Project administration:** Yukimasa Shibata.

**Supervision:** Yukimasa Shibata, Kiyoji Nishiwaki.

**Writing – original draft:** Yukimasa Shibata, Kiyoji Nishiwaki.

**Writing – review & editing:** Yukimasa Shibata, Kiyoji Nishiwaki.

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
