## [Decision Letter · Decision Letter 0]

9 Apr 2024

PONE-D-24-09277Mutations in fibulin-1 and collagen IV suppress the short healthspan of mig-17/ADAMTS mutants in Caenorhabditis elegansPLOS ONE

Dear Dr. Shibata,

Thank you for submitting your manuscript to PLOS ONE. After careful consideration, we feel that it has merit but does not fully meet PLOS ONE’s publication criteria as it currently stands. Therefore, we invite you to submit a revised version of the manuscript that addresses the points raised during the review process. While reviewers think the work is interesting, both felt that the manuscript needs improvements. Please refer to the reviewer's comments. In the revised manuscript, attach a letter addressing reviewer's critiques point-by-point when submitting the revised manusript. 

We look forward to receiving your revised manuscript.

Kind regards,

Myeongwoo Lee, Ph.D.

Academic Editor

PLOS ONE

Journal Requirements:

"Grant-in-Aid for Research Activity Start-up by the Ministry of Education, Culture, Sports, Science and Technology to YS(22K20658) and by the Naito Grant for the advancement of natural science to KN. "

"We thank Noriko Nakagawa, Nami Okahashi, and Chizu Yoshikata for technical assistance. Some nematode strains used in this work were provided by the Caenorhabditis Genetics Center, which is funded by the National Institutes of Health National Center for Research Resources. This work was supported by a Grant-in-Aid for Research Activity Start-up by the Ministry of Education, Culture, Sports, Science and Technology to YS(22K20658) and by the Naito Grant for the advancement of natural science to KN."

"Grant-in-Aid for Research Activity Start-up by the Ministry of Education, Culture, Sports, Science and Technology to YS(22K20658) and by the Naito Grant for the advancement of natural science to KN. "

Reviewers' comments:

Reviewer's Responses to Questions

**Comments to the Author**

1. Is the manuscript technically sound, and do the data support the conclusions?

Reviewer #1: Yes

Reviewer #2: Partly

2. Has the statistical analysis been performed appropriately and rigorously? 

Reviewer #1: No

Reviewer #2: Yes

3. Have the authors made all data underlying the findings in their manuscript fully available?

Reviewer #1: Yes

Reviewer #2: Yes

4. Is the manuscript presented in an intelligible fashion and written in standard English?

Reviewer #1: Yes

Reviewer #2: Yes

5. Review Comments to the Author

Reviewer #1: This paper from a leader in the C. elegans organogenesis field describes how matrix proteins contribute to organismal aging. In addition to its role in DTC migration, the metalloprotease MIG-17/ADAMTS is important for healthy aging in the worm. Two other matrix proteins, LET-2/collagen and FBL-1/fibulin that interact genetically with MIG-17 in DTC migration also suppress age-related phenotypes in the mig-17 mutants in an allele-specific manner. I think this will be of interest to those interested in matrix and aging.

Recommendations:

Make it more clear in the abstract that let-2 and fbl-1 differentially affect the lifespan of mig-17 mutants, and that mig-17 shows age related phenotypes (say what these are) but not short lifespan per se. It might be more correct to say that mig-17 shows altered healthspan, not altered lifespan.

Provide a bit more information in the introduction about the roles of basement membrane in aging. What is known about this? (e.g. skin aging and collagen). This will make the article more accessible to a broader audience. Also provide a brief general description of the mig-17 age-dependent phenotypes.

Methods: Please provide more detail about the imaging (DIC, I assume?).

Statistics: Please also include a section describing the statistical methods. Each figure legend should more clearly describe the statistical methods used (just saying t-test is not sufficiently descriptive). Multiple comparisons corrections should be applied.

Include the supplementary diagram showing the position of the alleles on the proteins in the main text. Include MIG-17 too, and show that the allele studied is a null. This should be figure 1. Also a diagram of the worm and/or a photo showing where these proteins are localized might help.

Fig. 4 show all data as in Fig. 3. Not clear why there are asterisks above and below the columns. Effects seem very minor – check for multiple comparisons correction in statistics.

Fig. S1 % abnormal animals – abnormal in what way? Gonad morphogenesis defects would be more descriptive.

Minor:

p.5 Change ‘In addition to its plays a role in gonadogenesis, MIG-17 also function in regulating healthy aging’ to ‘In addition to its role in in gonadogenesis, MIG-17 also functions in regulating healthy aging’.

p. 6 ‘young adult animals were collected as day 1 adults’ and ‘number of pumps during 30 seconds’ ‘Average of three independent measurements were calculated for each individual’

p. 7 Add a period after ‘punctuated by molting’.

p.9 ‘Additionally we investigated changes in body length during aging in this study’. This sentence seems out of place. Recommend delete and just merge the previous sentence in with the next paragraph.

p. 9 last sentence Delete ‘were observed to’. It is fine to just say ‘The pumping rates decreased in day 9 adults’. Better would be to provide the quantification in the text. E.g decreased by 20% or whatever it is.

p.15 ‘Fibulin-1 IS also known to regulate…’

Fig. 1 Change ‘hatch’ to ‘hatching’ . T-tests not the appropriate stat for comparing percentages. Recommend using Chi-squared or Fisher’s exact.

All figures: X axis fonts are very tiny. Turning these to an angle might help.

Fig. 3 Y axis: Number of pumps in 30 sec

Fig. 5 fonts way too tiny

Fig. 6 should be Table 1.

Reviewer #2: This manuscript describes the genetic interaction between mig-17 and fbl-1 or let-2 viable alleles. The authors argue that allele-specific suppression of age-related phenotypes of the mig-17 null allele is present. This paper is an extension of their work published previously. This manuscript only displays behavioral analyses without cell biological data. However, their studies can provide important insight. The remodeling of the matrix may affect an animal's age-related markers.

While it contains exciting studies, the manuscript requires improvements to be published in PLOS One.

1. Introduction. The author should summarize how cell-matrix interaction links to age-related phenotypes, including examples from other systems.

2. Materials and Methods. The authors must include details of procedures (p.6). ‘Body length was measured using ImageJ software’; I suggest adding more detail. “ … the growth was assessed on the stage of vulva development …” also needs more detail.

3. Age-related phenotype suppression (Figure 5, difficult to read). I noticed that the median life span of mig-17 in panels A, B, and C ranged from 18 to 21 days, but the median of mig-17 in panel D appears to be around 16 days. The authors need to clarify the differences. Or, the experiment must be repeated.

4. Lifespan phenotypes of mutants. The authors described the reduced lifespan. In general, any mutations can reduce lifespan. Although the genetic study is interesting, it is difficult to convince that such mutations directly affect lifespan.

5. Minor comments.

a. Some typos need to be corrected. Ex) 'let-2(196)' and 'TGFβ'.

b. Graphs are difficult to understand. Figure 5 graph must be drawn using thicker lines.

c. Overall, the manuscript needs to be revised to a cohesive format.

6. PLOS authors have the option to publish the peer review history of their article (what does this mean?). If published, this will include your full peer review and any attached files.

Reviewer #1: **Yes: **Erin J. Cram

Reviewer #2: No

---

## [Author Response · Author response to Decision Letter 0]

3 May 2024

Followings are responses to editor’s and reviewer’s comment.

Comments to the Author

Reviewer #1: This paper from a leader in the C. elegans organogenesis field describes how matrix proteins contribute to organismal aging. In addition to its role in DTC migration, the metalloprotease MIG-17/ADAMTS is important for healthy aging in the worm. Two other matrix proteins, LET-2/collagen and FBL-1/fibulin that interact genetically with MIG-17 in DTC migration also suppress age-related phenotypes in the mig-17 mutants in an allele-specific manner. I think this will be of interest to those interested in matrix and aging.

Recommendations:

1. Make it more clear in the abstract that let-2 and fbl-1 differentially affect the lifespan of mig-17 mutants, and that mig-17 shows age related phenotypes (say what these are) but not short lifespan per se. It might be more correct to say that mig-17 shows altered healthspan, not altered lifespan.

 We added underlined phrase in the abstract.

Page 3, line 11

“Interestingly, fbl-1(k206), but not fbl-1(k201) or let-2 alleles, exhibited an extended lifespan compared to the wild type when combined with mig-17.”

Page 3, line 14

“In addition to the control of DTC migration, MIG-17 also plays a role in healthspan, but not in lifespan”

2. Provide a bit more information in the introduction about the roles of basement membrane in aging. What is known about this? (e.g. skin aging and collagen). This will make the article more accessible to a broader audience. Also provide a brief general description of the mig-17 age-dependent phenotypes.

We added following sentences to describe the roles of basement membrane in aging.

Page 4, line2

“The BMs are complex structures composed of multiple types of molecules. To maintain their integrity, it is necessary to appropriately remove damaged components and incorporate new ones, ensuring proper turnover. However, as animals age, the metabolism of BMs slows down, leading to the accumulation of damage [1]. Collagen levels decrease with age, while spontaneous cross-links increase. Yet, the anti-aging effects of BMs remain poorly understood. In the skin, BM collagen is essential for stem cell maintenance through hemidesmosomes [2]. Loss of collagen is known to impair cell competition, thus contributing to aging. However, the relationship between BMs and the suppression of aging beyond skin stem cells remains unknown.”

Accordingly, we added two references

1. Ewald CY. The Matrisome during Aging and Longevity: A Systems-Level Approach toward Defining Matreotypes Promoting Healthy Aging. Gerontology. 2020;66(3):266-74. Epub 20191213. doi: 10.1159/000504295. PubMed PMID: 31838471; PubMed Central PMCID: PMCPMC7214094.

2. Liu N, Matsumura H, Kato T, Ichinose S, Takada A, Namiki T, et al. Stem cell competition orchestrates skin homeostasis and ageing. Nature. 2019;568(7752):344-50. Epub 20190403. doi: 10.1038/s41586-019-1085-7. PubMed PMID: 30944469.

To describe the mig-17 age-dependent phenotypes, we modified the description as follows.

Page 5, 3rd line from the bottom

“mig-17 mutants exhibit an earlier decline in periodic behavior including the defecation cycle and pumping rate, and motility rate, as well as earlier shortening of body length compared to the wild type”

3. Methods: Please provide more detail about the imaging (DIC, I assume?).

We added underlined phrase.

Page 6, 5th line from the bottom

“Gonad migration phenotypes were scored using a DIC images of Nomarski microscope (Axioplan 2; Zeiss).”

4. Statistics: Please also include a section describing the statistical methods. Each figure legend should more clearly describe the statistical methods used (just saying t-test is not sufficiently descriptive). Multiple comparisons corrections should be applied.

Thank you for your suggestion. We performed multiple comparisons corrections in Fig. 3, 4, 5 using one-way ANOVA and Tukey’s multiple comparisons. 

Accordingly, we changed figure legends as follows.

Legend of figure 3-5

We removed “for t-test” from following sentence “p-values for t-test against WT is indicate at the bottom of plots.”

We replaced“: *** p<0.005, ** p<0.01, *p<0.05.” to “One-way ANOVA with Tukey’s multiple comparisons:, *p≤ 0.05, **p≤ 0.01.”

We added statistics section in Materials and Methods.

Page 8, line 6

“Statistics

Growth rates were analyzed using Fisher's exact test to compare adults and larvae.

One-way analysis of variance (ANOVA) was conducted using Excel to compare defecation cycle, pumping rate, and body length. If a significant difference was found by ANOVA, Tukey’s multiple comparison test was performed using Excel statistics.

The lifespan was analyzed using a log-rank test in R.”

Accordingly, we changed result and discussion as follows.

Page 10, 4th line from the bottom 

“fbl-1(k206) alleles showed significantly longer defecation cycles compared to the wild type after day 5 and fbl-1(k201) showed a trend toward suppression of the mig-17 defecation decline at day 5, although it was not statistically significant (t-test: p=0.069)”

Page 11, line 4

“The pumping rates of mig-17 adults were much slower than those of wild type at day 1 and day 9, suggesting short healthspan of mig-17 animals (Fig. 4C, D) [11]. Interestingly, let-2(k196), but not let-2(k193), showed a trend toward suppression of the mig-17 pumping rate decline at day 9 (Fig. 4A, B), although it was not statistically significant. The pumping rate of mig-17(k174) and fbl-1(k206); mig-17(k174), but not that of fbl-1(k201); mig- 17(k174), showed significant differences from that of wild-type animals at day 9 adult. (Fig. 4C, D).”

Page 11, 6th line from the bottom 

“The body length of day 1 adults was longer in mig-17; let-2(k196) and mig-17; let-2(k193) double mutants compared to the wild type or mig-17 mutants. At the day3 adult, the body length of mig-17(k174) and let-2(k193) mig-17(k174), but not that of let-2(k196) mig- 17(k174), showed significant differences from those of wild-type animals”

Page 14, line 8

 “However, let-2(k196) suppressed the early prolongation of mig-17 defecation cycle associated with aging.”

We also changed Table1 as follows.

Defecation cycle: mig-17(k174) fbl-1(k201) and fbl-1(k201) became blank.

Pumping rate: mig-17(k174) let-2(k193), let-2(k196), and let-2(k193) became blank.

Body length (represent day3 data): mig-17(k174) fbl-1(k206) and fbl-1(k206) became short.

5. Include the supplementary diagram showing the position of the alleles on the proteins in the main text. Include MIG-17 too, and show that the allele studied is a null. This should be figure 1. Also a diagram of the worm and/or a photo showing where these proteins are localized might help.

We added Figure 1, which includes a diagram showing the position of the alleles on the MIG-17, LET-2, and FBL-1 proteins. Additionally, we included a diagram indicating the localization of these proteins on BMs. 

Accordingly, we added legend of Figure 1, as follows.

“Figure 1: Schematic diagrams of MIG-17, FBL-1C, and LET-2A.

(A) The mutation sites of mig-17(k174), let-2(k193, k196), and fbl-1(k201, k206) are indicated. (B) A diagram showing the localization of MIG-17, LET-2, and FBL-1. These proteins are components of BMs.

”

And we removed the description of alleles of FBL-1 and LET-2 from Sup Figure1.

Accordingly, we changed Fig. 1-5 to Fig. 2-6 and removed Sup Figure1A.

6. Fig. 4 show all data as in Fig. 3. Not clear why there are asterisks above and below the columns. Effects seem very minor – check for multiple comparisons correction in statistics.

Thank you for your suggestion. We represented Fig. 5(ex-Fig. 4) as box and dot plot.

Accordingly, we changed first sentence of legend as follows.

“Box and dot plots indicate the body length.”

7. Fig. S1 % abnormal animals – abnormal in what way? Gonad morphogenesis defects would be more descriptive.

We changed description of y axis to “% animals w gonad morphogenesis defects”

Minor:

p.5 Change ‘In addition to its plays a role in gonadogenesis, MIG-17 also function in regulating healthy aging’ to ‘In addition to its role in in gonadogenesis, MIG-17 also functions in regulating healthy aging’.

We followed reviewer’s suggestion.

p. 6 ‘young adult animals were collected as day 1 adults’ and ‘number of pumps during 30 seconds’ ‘Average of three independent measurements were calculated for each individual’

We followed reviewer’s suggestion.

p. 7 Add a period after ‘punctuated by molting’.

We followed reviewer’s suggestion.

p.9 ‘Additionally we investigated changes in body length during aging in this study’. This sentence seems out of place. Recommend delete and just merge the previous sentence in with the next paragraph.

We followed reviewer’s suggestion.

p. 9 last sentence Delete ‘were observed to’. It is fine to just say ‘The pumping rates decreased in day 9 adults’. Better would be to provide the quantification in the text. E.g decreased by 20% or whatever it is.

We changed to “The pumping rates decreased by 50% in day 9 adults.” (Page 10, 1st line)

p.15 ‘Fibulin-1 is also known to regulate…’

We followed reviewer’s suggestion.

Fig. 1 Change ‘hatch’ to ‘hatching’ . T-tests not the appropriate stat for comparing percentages. Recommend using Chi-squared or Fisher’s exact.

We followed reviewer’s suggestion.

Accordingly, we changed figure legend as follows.

“Black and red asterisks on the top of bar graph indicate p-values for Fisher’s exact test against WT and mig-17(k174), respectively”

In addition, we performed Fisher’s exact test to Supplementary data that analyzed suppression of the gonad migration defect of mig-17.

All figures: X axis fonts are very tiny. Turning these to an angle might help.

We followed reviewer’s suggestion.

Fig. 3 Y axis: Number of pumps in 30 sec

We followed reviewer’s suggestion.

Fig. 5 fonts way too tiny

We followed reviewer’s suggestion.

Fig. 6 should be Table 1.

We followed reviewer’s suggestion.

Reviewer #2: This manuscript describes the genetic interaction between mig-17 and fbl-1 or let-2 viable alleles. The authors argue that allele-specific suppression of age-related phenotypes of the mig-17 null allele is present. This paper is an extension of their work published previously. This manuscript only displays behavioral analyses without cell biological data. However, their studies can provide important insight. The remodeling of the matrix may affect an animal's age-related markers.

While it contains exciting studies, the manuscript requires improvements to be published in PLOS One.

1. Introduction. The author should summarize how cell-matrix interaction links to age-related phenotypes, including examples from other systems.

Please see response to reviwer1 point 2.

2. Materials and Methods. The authors must include details of procedures (p.6). ‘Body length was measured using ImageJ software’; I suggest adding more detail. “ … the growth was assessed on the stage of vulva development …” also needs more detail.

We added following sentences to explain details of procedures.

Page 7, line 8

“To measure body length, we traced a line along the body using ImageJ and then measured its length.”

Page 7, line 2

“A simple invagination occurs in early L4, followed by a Christmas tree-like invagination in mid-L4, and the invagination becomes mostly closed in late L4.”

3. Age-related phenotype suppression (Figure 5, difficult to read). I noticed that the median life span of mig-17 in panels A, B, and C ranged from 18 to 21 days, but the median of mig-17 in panel D appears to be around 16 days. The authors need to clarify the differences. Or, the experiment must be repeated.

C. elegans lifespans vary from experiment to experiment, both for wild type and mutants. This variability is believed to be influenced by multiple factors, including the degree of dryness of the plate and the amount of E. coli used as food. Given the difficulty in controlling these conditions, when measuring the lifespan of C. elegans, we compare the control and sample using plates seeded with E. coli prepared simultaneously. Discrepancies in lifespan may arise if plates are created at different times. The variance in the lifespan of mig-17 between experiments falls within the normal range. Moreover, in both instances, the results are reproducible, as there are no significant differences observed between the wild type and mig-17 mutants.

4. Lifespan phenotypes of mutants. The authors described the reduced lifespan. In general, any mutations can reduce lifespan. Although the genetic study is interesting, it is difficult to convince that such mutations directly affect lifespan.

We agree with reviewer’s suggestion. Therefore, we added underlined phrase in discussion part.

Page 14, line 6

“Thus, both of these type IV collagen mutations have a negative impact on lifespan, although it is not clear whether these mutations directly affect lifespan.”

5. Minor comments.

a. Some typos need to be corrected. Ex) 'let-2(196)' and 'TGFβ'.

We corrected these typo.

b. Graphs are difficult to understand. Figure 5 graph must be drawn using thicker lines.

We followed reviewer’s suggestion.

c. Overall, the manuscript needs to be revised to a cohesive format.

We followed reviewer’s suggestion.

---

## [Decision Letter · Decision Letter 1]

20 May 2024

PONE-D-24-09277R1Mutations in fibulin-1 and collagen IV suppress the short healthspan of mig-17/ADAMTS mutants in Caenorhabditis elegansPLOS ONE

Dear Dr. Shibata,

Thank you for submitting your manuscript to PLOS ONE. After careful consideration, we feel that it has merit but does not fully meet PLOS ONE’s publication criteria as it currently stands. Therefore, we invite you to submit a revised version of the manuscript that addresses the points raised during the review process.

Reviewers only require minor modifications and calrifications, which can be completed easily.

We look forward to receiving your revised manuscript.

Kind regards,

Myeongwoo Lee, Ph.D.

Academic Editor

PLOS ONE

Journal Requirements:

Reviewers' comments:

Reviewer's Responses to Questions

**Comments to the Author**

1. If the authors have adequately addressed your comments raised in a previous round of review and you feel that this manuscript is now acceptable for publication, you may indicate that here to bypass the “Comments to the Author” section, enter your conflict of interest statement in the “Confidential to Editor” section, and submit your "Accept" recommendation.

Reviewer #1: All comments have been addressed

Reviewer #2: All comments have been addressed

2. Is the manuscript technically sound, and do the data support the conclusions?

Reviewer #1: Yes

Reviewer #2: Yes

3. Has the statistical analysis been performed appropriately and rigorously? 

Reviewer #1: Yes

Reviewer #2: No

4. Have the authors made all data underlying the findings in their manuscript fully available?

Reviewer #1: Yes

Reviewer #2: Yes

5. Is the manuscript presented in an intelligible fashion and written in standard English?

Reviewer #1: Yes

Reviewer #2: Yes

6. Review Comments to the Author

Reviewer #1: Change the Microscopy section in the materials and methods to:

Gonad migration phenotypes were scored using DIC images captured with a Zeiss Axioplan 2

microscope.

Under ‘Behavioral Analysis’, recommend edits to make complete sentences: ‘Number of pumps

during 30 seconds were counted and the average of three independent measurements was calculated for each

individual.

Reviewer #2: 1. Please clarify the purpose of Figure 1. The domain structures were not mentioned in the Introduction. Specifically, Figure 1B should have been mentioned in the text. I suggest considering its deletion.

2. Materials and methods. References are needed for the statistical methods used in the study.

3. Page 9, line 9. “the gain of function mutations in collagen IV..” needs a reference.

4. Page 11, line 7. “.. showed significant differences ..” needs a p-value in parenthesis.

7. PLOS authors have the option to publish the peer review history of their article (what does this mean?). If published, this will include your full peer review and any attached files.

Reviewer #1: No

Reviewer #2: No

---

## [Author Response · Author response to Decision Letter 1]

22 May 2024

Comments to the Author

Reviewer #1: Change the Microscopy section in the materials and methods to:

Gonad migration phenotypes were scored using DIC images captured with a Zeiss Axioplan 2 microscope.

Thank you for your suggestion. We followed reviewer’s comment.

Under ‘Behavioral Analysis’, recommend edits to make complete sentences: ‘Number of pumps during 30 seconds were counted and the average of three independent measurements was calculated for each individual.

We followed reviewer’s comment.

Reviewer #2: 1. Please clarify the purpose of Figure 1. The domain structures were not mentioned in the Introduction. Specifically, Figure 1B should have been mentioned in the text. I suggest considering its deletion.

We cited Figure 1A, and 1B in Introduction as follows.

Page 5 line2

“MIG-17 is secreted from the body wall muscle cells as a proform and localizes to the BMs of various tissues, including gonad, intestine, body wall muscle, and hypodermis, where it is activated by auto-catalytic removal of its prodomain [5, 6] (Fig. 1A, B).”

Page 5 line7

“Dominant gain-of-function (gf) mutations in the BM molecules collagen IV �2 chain and fibulin-1 suppress the gonadal defect of the mig-17 mutants [8-10] (Fig. 1A). let-2(gf) mutations are amino-acid substitutions either in the triple helical domain or in the C-terminal non-collagenous 1 (NC1) domain (Fig. 1A). fbl-1(gf) mutations are amino-acid substitutions found in the second EGF-like motif (Fig. 1A).”

2. Materials and methods. References are needed for the statistical methods used in the study.

Thank you for your suggestion. We followed reviewer’s comment.

Followings are references.

Shibata Y, Tanaka Y, Sasakura H, Morioka Y, Sassa T, Fujii S, et al. Endogenous chondroitin extends the lifespan and healthspan in C. elegans. Sci Rep. 2024;14(1):4813. Epub 20240227. doi: 10.1038/s41598-024-55417-7. PubMed PMID: 38413743; PubMed Central PMCID: PMCPMC10899230.

Choi W, Ryu SE, Cheon Y, Park YJ, Kim S, Kim E, et al. A single chemosensory GPCR is required for a concentration-dependent behavioral switching in C. elegans. Curr Biol. 2022;32(2):398-411 e4. Epub 20211213. doi: 10.1016/j.cub.2021.11.035. PubMed PMID: 34906353.

3. Page 9, line 9. “the gain of function mutations in collagen IV..” needs a reference.

We followed reviewer’s comment.

Followings are references.

Kubota Y, Kuroki R, Nishiwaki K. A fibulin-1 homolog interacts with an ADAM protease that controls cell migration in C. elegans. Curr Biol. 2004;14(22):2011-8. doi: 10.1016/j.cub.2004.10.047. PubMed PMID: 15556863.

Imanishi A, Aoki Y, Kakehi M, Mori S, Takano T, Kubota Y, et al. Genetic interactions among ADAMTS metalloproteases and basement membrane molecules in cell migration in Caenorhabditis elegans. PLoS One. 2020;15(12):e0240571. doi: 10.1371/journal.pone.0240571. PubMed PMID: 33264296; PubMed Central PMCID: PMCPMC7710118.

4. Page 11, line 7. “.. showed significant differences ..” needs a p-value in parenthesis.

We added p-value as follows.

Page 11, line 7

“The pumping rate of mig-17(k174) (p=0.0001) and fbl-1(k206) mig-17(k174) (p=0.028), but not that of fbl-1(k201); mig-17(k174) (p=0.105), showed significant differences from that of wild-type animals at day 9 adult (Fig. 4C, D).”

---

## [Decision Letter · Decision Letter 2]

30 May 2024

Mutations in fibulin-1 and collagen IV suppress the short healthspan of mig-17/ADAMTS mutants in Caenorhabditis elegans

PONE-D-24-09277R2

Dear Dr. Shibata,

We’re pleased to inform you that your manuscript has been judged scientifically suitable for publication and will be formally accepted for publication once it meets all outstanding technical requirements.

Kind regards,

Myeongwoo Lee, Ph.D.

Academic Editor

PLOS ONE

Additional Editor Comments (optional):

Reviewers' comments:

Reviewer's Responses to Questions

**Comments to the Author**

1. If the authors have adequately addressed your comments raised in a previous round of review and you feel that this manuscript is now acceptable for publication, you may indicate that here to bypass the “Comments to the Author” section, enter your conflict of interest statement in the “Confidential to Editor” section, and submit your "Accept" recommendation.

Reviewer #1: All comments have been addressed

Reviewer #2: All comments have been addressed

2. Is the manuscript technically sound, and do the data support the conclusions?

Reviewer #1: Yes

Reviewer #2: Yes

3. Has the statistical analysis been performed appropriately and rigorously? 

Reviewer #1: Yes

Reviewer #2: I Don't Know

4. Have the authors made all data underlying the findings in their manuscript fully available?

Reviewer #1: Yes

Reviewer #2: Yes

5. Is the manuscript presented in an intelligible fashion and written in standard English?

Reviewer #1: Yes

Reviewer #2: Yes

6. Review Comments to the Author

Reviewer #1: The authors have addressed all of my concerns. I have no further comments on the paper, which is a nice addition to the cell migration literature.

Reviewer #2: No further comments, except for Figure 1B. I still don't understand why the authors need Figure 1B. It does not add anything to understand their model.

7. PLOS authors have the option to publish the peer review history of their article (what does this mean?). If published, this will include your full peer review and any attached files.

Reviewer #1: No

Reviewer #2: No

---

## [Editor Report · Acceptance letter]

4 Jun 2024

PONE-D-24-09277R2 

PLOS ONE

Dear Dr. Shibata, 

I'm pleased to inform you that your manuscript has been deemed suitable for publication in PLOS ONE. Congratulations! Your manuscript is now being handed over to our production team.

Kind regards, 

on behalf of

Dr. Myeongwoo Lee 

Academic Editor

PLOS ONE